Effects of grazing strategy on facultative grassland bird nesting on native grassland pastures of the Mid-South USA

Buckley Byron R. brbuckley@agcenter.lsu.edu nwtf01@gmail.com 1 2
Lituma Christopher M. 1 3
Keyser Patrick D. 1
Holcomb Elizabeth D. 4
Smith Ray 5
Morgan John J. 6
Applegate Roger D. 1 7
1 University of Tennessee - Knoxville , Knoxville , TN , United States of America
2 Louisiana State University Agricultural Center , Baton Rouge , LA , United States of America
3 West Virginia University , Morgantown , WV , United States of America
4 University of Tennessee - Knoxville , Knoxville , United States of America
5 University of Kentucky , Lexington , KY , United States of America
6 National Bobwhite Conservation Initiative , Lawrenceburg , KY , United States of America
7 Tennessee Wildlife Resources Agency , Nashville , TN , United States of America
Kramer Donald
Electronic publication date: 2022 Sep 28
Publication date: 2022
Volume: 10
Electronic Location ID: e13968
Received 2022 Apr 26; Accepted 2022 Aug 8
Copyright: ©2022 Buckley et al.
Copyright year: 2022
Copyright holder: Buckley et al.
License: This is an open access article distributed under the terms of the Creative Commons Attribution License, which permits unrestricted use, distribution, reproduction and adaptation in any medium and for any purpose provided that it is properly attributed. For attribution, the original author(s), title, publication source (PeerJ) and either DOI or URL of the article must be cited.
License URL: https://creativecommons.org/licenses/by/4.0/

Keywords: Grassland birds, Native warm-season grasses, Nesting, Nest-site selection, Patch-burn grazing

Funding: Conservation Innovation Grant program 69-3A75-11-176 Tennessee Hatch Project TEN00463 and TEN00547 This work was funded by the USDA-NRCS through the Conservation Innovation Grant program (award number 69-3A75-11-176), BASF, USDA, and the University of Tennessee Institute of Agriculture. The funders had no role in study design, data collection and analysis, decision to publish, or preparation of the manuscript.

==============================
Understanding how livestock grazing strategies of native warm season grasses (NWSG) can impact facultative grassland bird nesting can provide insight for conservation efforts. We compared pre and post treatment effects of rotational grazing (ROT) and patch-burn grazing (PBG) for facultative grassland bird species nest success and nest-site selection on NWSG pastures at three Mid-South research sites. We established 14, 9.7 ha NWSG pastures and randomly assigned each to either ROT or PBG and monitored avian nest-site selection and nest success, 2014–2016. We collected nesting and vegetation data in 2014, before treatment implementation, as an experimental pre-treatment. We implemented treatments across all research sites in spring 2015. We used a step-wise model selection framework to estimate treatment effect for ROT or PBG on avian nest daily survival rate (DSR) and resource selection function (RSF) at the temporal scale and within-field variables. Daily survival rates were 0.93% (SE = 0.006) for field sparrow (Spizella pusilla), 0.96% (SE = 0.008) for red-winged blackbird (Agelaius phoeniceus), and 0.92% (SE = 0.01) for indigo bunting (Passerina cyanea). Model support for PBG treatment and vegetation height were indicated as negative and positive influences for field sparrow DSR, respectively. Red-winged blackbirds’ DSR were negatively influenced by ROT while vegetation height positively affected DSR, and DSR for indigo bunting did not differ among treatments. Combined RSF models indicated nest-site selection for all species was positively related to vegetation height and only weakly associated with other within-field variables. We provide evidence that ROT and/or PBG effects vary by species for DSR for these three facultative grassland birds, and vegetation characteristics affected their nest-site selection in the Mid-South USA. A lack of disturbance in Mid-South grasslands can lead to higher successional stages (i.e., mix shrub-grassland), but some combination of ROT, PBG, and unburned/ungrazed areas can provide adequate nesting habitat on small pasture lands (∼1.8 –7.8 ha) for various facultative grassland birds and potentially offer the opportunity to simultaneously maintain livestock production and grassland bird nesting habitat.

Introduction

Grassland bird populations in North America have experienced a ∼45% decline since the 1970s (Rosenberg et al., 2019). Habitat loss, fragmentation, and degradation through fire suppression, and inappropriate grazing management are contributing causes of these declines (Green et al., 2005; White, Murray & Rohweder, 2000). Much of the eastern United States has experienced reforestation due to fire suppression, which has also reduced grassland habitat. Remaining grasslands within the eastern United States have been converted to non-native grass species (tall fescue (Schedonorus arundinaceus (Schreb.) Dumort.), orchard grass (Dacytlis glomerata—L.), bermudagrass (Cynodon dactylon—(L.) Pers.)) focused primarily on livestock production (Derner et al., 2009; Tilman, 1999). These conversions have led to alterations in vegetative structure and composition of Mid-South (the region south of glacial influence, north of the Gulf Coastal Plain, west of the Appalachians, and east of the Great Plains (Barrioz et al., 2013)) grasslands (Van Auken, 2000; Briske et al., 2011; Hayes & Holl, 2003; Willcox et al., 2010). In turn, these changes have been linked to reduced nesting success and shifts in nest-site selection for grassland bird populations (Coppedge et al., 2008; Davis, 2005; Herakovich et al., 2021; Roberts, Boal & Whitlaw, 2017).

To mitigate the loss of grassland habitat at a large scale, “working-lands conservation” efforts promote sustainable grazing practices on private lands to benefit agricultural production and grassland bird populations (Keyser et al., 2019; Kremen & Merenlender, 2018; Monroe et al., 2016). Under a working-lands model, native warm-season grass pastures managed with rotational grazing (ROT) or patch-burn grazing (PBG) could improve grassland bird breeding habitat and contribute to conservation efforts in eastern systems (Lituma et al., 2022). Grazing management that relies on the systematic shifting of cattle (Bos tarus) at temporal and spatial scales (ROT) can achieve uniform utilization of forage within a given pasture while creating heterogeneous vegetation structure among pastures (Briske et al., 2011; Holling, 1978). However, research comparing ROT with other land management strategies (i.e., continuously grazed) has produced variable results concerning grassland bird nesting success. Studies have reported reduced nest success for facultative grassland bird species (utilize grassland habitat as a part of a wider array of habitats) for ROT versus idle or continuously grazed pastures (Temple et al., 1999). This conflicting information suggests that nest success under ROT is species and/or region-specific for facultative grassland birds. Use of ROT also indicates that impacts on structure and plant species composition will determine benefits among facultative grassland birds (Sliwinski, Powell & Schacht, 2019; Soderstrom, Part & Linnarsson, 2001; Temple et al., 1999).

Pyric-herbivory (i.e., periodic fires and large ungulate grazing), is hypothesized to mimic the historical natural disturbances under which North American grassland ecosystems evolved (Fuhlendorf et al., 2009). Grazing management based on pyric herbivory (i.e., PBG), utilizes prescribed burns to create a mosaic of burned and unburned areas across a gradient of spatial and temporal scales within grasslands (Fuhlendorf et al., 2009). Selective grazing of recently burned areas results in increased vegetation structural and compositional heterogeneity (Allred et al., 2011; Augustine & Derner, 2015). Researchers have reported similar (Holcomb, Davis & Fuhlendorf, 2014), and highly variable (Doxon, 2009) facultative grassland bird nest success when compared to traditional grazing. Given this uncertainty it is important to examine PBG disturbance effects given the highly variable response for facultative grassland bird nest survival and the lack of empirical data in the eastern USA.

It is imperative to analyze ROT and PBG management practices across ecosystems (i.e., semi-arid grasslands of the Great Plains and humid, temperate Mid-South grasslands) due to variation in landscape context, precipitation gradients, and bird species-specific responses. Much of the current ROT/PBG peer-reviewed literature originates from the semi-arid Great Plains ecosystem. Furthermore, a direct comparison between ROT and PBG and their effects on grassland birds is needed in the Mid-South USA. In north Mississippi’s Black Belt Prairie, ROT management was used to promote NWSG, which resulted in higher nest density for dickcissels due to the increase in habitat structural heterogeneity (Conover, Dinsmore & Burger, 2011; Monroe et al., 2016). Conversely, Harper et al. (2015) found that full-season grazing (early May to late summer) would maintain favorable vegetation structure (vegetation height average pasture = ∼40 cm) suitable for grassland birds nesting and brooding habitat in Tennessee. The utilization of PBG in the Mid-South USA could potentially improve grassland bird populations on working lands (Keyser et al., 2019) or, at minimum, provide nesting habitat without sacrificing cattle production.

Understanding the efficacy of ROT and PBG native grassland management on bird reproductive potential can aid working-lands conservation in pasturelands of the Mid-South USA and potentially inform conservation strategies in other regions. Therefore, we evaluated ROT and PBG effects on vegetation characteristics at the within-field scale to determine if these grazing strategies affect grassland avian species reproductive success (DSR and nest success) and nest-site selection on NWSG pastures in the humid temperate Mid-South United States. Additionally, we assessed if grassland bird reproductive efforts during a pre-treatment year (ungrazed and unburned) on the same pastures were affected by subsequent treatments. Finally, we examined the influence of within-field vegetation characteristics (structure and composition) on DSR and nest-site selection. We hypothesized that PBG pastures would provide more favorable vegetation characteristics due to an increased heterogeneous structure at the within-field scale for facultative grassland birds resulting in greater reproductive success (DSR and nest success) and selection for nesting locations than pastures managed with ROT or ungrazed pre-treatments.

Material and Methods

Study area and site preparation

We conducted our research on three sites: (1) Blue Grass Army Depot (BGAD) in Madison County in east-central Kentucky (37°41′31″N, 84°10′56″W’; elevation, 283 m), (2) Quicksand, Robinson Center for Appalachian Resource Sustainability (QUICK) in Breathitt County in eastern Kentucky (37°25′42″N, 83°10′22″W; elevation, 383 m) and (3) Dairy Research and Education Center (DREC) in Marshall County in south-central Tennessee (35°24′58″N, 86°48′50″W; elevation, 251 m; Fig. 1). The BGAD and DREC sites were located in the Bluegrass and Highland Rim Section of the Interior Lower Plateau (Griffith, 2010; Interior Low Plateau Ecoregional Planning Team, 2005) while QUICK was located in the North Cumberland Plateau of the Southern Appalachian ecoregions (Griffith, 2010; Bullington & Wheaton, 2003). The Interior Lower Plateau consists of irregular plains, open hills, and smooth plains with an elevation between ∼200–300 m with an average annual precipitation of ∼111 cm. The Interior Lower Plateau is generally described as a predominately oak (Quercus spp.)-hickory (Cary spp.) forested region with sections of tallgrass prairie (Interior Low Plateau Ecoregional Planning Team, 2005). The North Cumberland Plateau is characterized by oak-hickory, oak-pine (Pinus spp.) mixed forest with agriculture pastures and reclaimed surface mines which range from ∼365–609 m in elevation and annual precipitation of ∼88–139 cm (Griffith, 2010; Bullington & Wheaton, 2003).

Figure 1 Study site locations used to examine livestock impacts on grassland-associated birds.

Study site location for patch-burn grazing and rotational grazing assessment of grassland-associated bird nest-site selection and nest success on native warm-season grasses pastures on three research sites in the Mid-South in Tennessee and Kentucky, USA from 2014–2016.

Pastures (9.7 ± 0.47 ha each) at each site were converted to NSWG from cool-season grasses during 2012–2013 (Keyser et al., 2015b). Stands were sown with a grass mixture that included 6.7 kg ha−1 (pure live seed basis) big bluestem (Andropogon gerardii [Vitman]), 3.3 kg ha−1 Indiangrass (Sorghastrum nutans [(L.) Nash]), and 1.1 kg ha−1 little bluestem (Schizachyrium scoparium [(Michx). Nash]). We established six pastures at BGAD, a property that also included tall fescue pastures, hayfields, and oak-dominated woodlots adjacent to NWSG pastures. We converted four pastures at DREC with similar land use as BGAD. At QUICK, we planted four pastures with the surrounding landscape being a reclaimed surface mine (reclaimed between 2004–2012) dominated by tall fescue, sericea lespedeza (Lespedeza cuneata [Dum. Cours.]) and stands of various planted hardwoods including autumn olive (Elaeagnus umbellate [Thunb.]) and American sycamore (Platanus occidentalis [Fer.]).

Treatments and management protocol

We used permanent fencing to create PBG and ROT pastures (n = 14) then used temporary fencing to divide ROT pastures into thirds (3.2 ha paddocks) for rotational grazing. We randomly selected half of the pastures at each site for PBG treatments and implemented prescribed burns on a different paddock each year, 2015–2016. We used ∼3 m disked lines as fire breaks around all burn pastures and all prescribed burns were conducted in early to mid-April of each burn year. Rotationally grazed pastures were not burned during this study. Pastures were not grazed or burned for either treatment during 2014 to allow them to complete establishment and to collect pre-treatment data.

We utilized an initial stocking density of cattle based on previous NWSG research in the Mid-South and adjusted rates across sites based on pasture conditions and site productivity (Keyser et al., 2015a). On the less productive mine site (QUICK), stocking density was 260–350 kg ha−1 while at BGAD it was 500–600 kg ha−1, and at DREC 620–700 kg ha−1. We stocked pastures ∼2–5 weeks post-burn for all sites. We used yearling heifers or, due to a lack of availability of heifers (QUICK only), steers for grazing purposes. Cattle grazed freely throughout PBG pastures. We rotated cattle on ROT pastures among the three paddocks based on residual vegetation height (target = 35–45 cm); in practice, we moved cattle approximately once every 4–7 days. We provided all cattle with water, shade, and trace mineral salt blocks for all pastures and across all three sites. Cattle occupied each pasture from mid-May until late August each year, 2015–2016. Animal care adhered to University of Tennessee-Institutional Animal Care and Use protocols No. 2258-0414 and No. 2258-0417.

Nest searching and monitoring

We searched for facultative grassland bird nests beginning from early May to late July across all research sites, 2014–2016. We located grassland bird nests using a combination of systematic point counts and behavioral observations of adults (Martin & Geupel, 2016; Winter et al., 2003). We searched each pasture every 3–4 days for potential grassland bird nests. Once a nest was located, we recorded the Universal Transverse Mercator (UTM) coordinates, species, parental activity, and nest contents (eggs or nestlings). We attached 10 cm orange vinyl flagging 5 m north of a nest to facilitate relocation. We monitored each nest every 2–3-days to determine fate (abandoned, successful, or failed nest) by recording the nest contents and parental activity. We categorized a successful nest as those with ≥1 nestling fledged. We determined fledging by observing parental behavior (i.e., adult alarm call and chick feeding calls) or visual confirmation of young near the nest (feces on the rim, flushed young near the nest). We determined a nest failure if eggs were missing, there were broken egg fragments in the nest, if behavioral cues (absent parents, absent fledglings) indicated failure, or the nest was destroyed.

Nest measurements

We collected vegetation measurements at all active nests within two weeks of completion (young fledged or failed). We measured nest substrate height (cm), nest height (measure to the rim of the cup; cm), litter depth (cm), and used a Daubenmire frame to estimate percent cover of grass, forbs, bare ground, and litter for each active nest location. We recorded visual obstruction (VOR) using a Robel pole in each cardinal direction (N, S, E, or W) 4 m from the center of each nest bowl (Robel et al., 1970).

Pasture vegetation measurements

We also conducted vegetation samples in each pasture during May, June, and July 2014–2016. We utilized previously established fixed avian point count locations, hereafter vegetation points, spaced >150 m apart within each pasture (n = ≤ 5 points/pasture). We measured within-field vegetation variables (the same ones previously mentioned for nest sites) along a 25 m transect in a randomly selected cardinal direction (Elzinga, Salzer & Willoughby, 1998), starting at each vegetation point center. Vegetation metrics were recorded every 5 m alternating between the left and right side of the transect line.

Statistical analysis

For data analysis we selected nests of those species that were of conservation concern (i.e., species listed on the Birds of Conservation Concern List; U.S. Fish and Wildlife Service Division of Migratory Bird Management, 2008), and had >30 nests (a number that permitted models to converge properly) (Moineddin, Matheson & Glazier, 2007; Smith et al., 1997). Before fitting models, we assessed vegetation measurement explanatory variables multicollinearity by calculating variance inflation factors (VIF) with the VIF function in the R package car, version 3.5.0 (Fox & Weisberg, 2018). We created a linear regression model with all vegetation measurement variables and removed variables with VIF values >5 (James et al., 2014). We used the nest survival model function built on a logistic regression framework in the RMark package in Program R (version 3.6.2 R Core Team, 2019) to estimate DSR for selected facultative grassland bird nests (Dinsmore, White & Knopf, 2002; Laake, 2013; White & Burnham, 1999). We grouped nests by species across all sites to increase sample size. We used a step-wise modeling approach (Mundry & Nunn, 2009; Whittingham et al., 2006) to determine the influence of site, treatment, and/or within-field variable effects on DSR for each selected facultative grassland bird species individually. We used Akaike’s Information Criterion corrected for small sample sizes (AICc) to evaluate model performance and identify competitive models (≤ 2.0 AICc) (Anderson, 2008; Burnham & Anderson, 2002). We considered variables with β-values with a 95% confidence interval that did not overlap zero to be important in explaining the variability in top models (Arnold, 2010). We created model subsets for DSR with an additive step-wise process, by modeling (1) year, (2) research site, (3) treatment method, and (4) within-field variables as covariates for each selected bird species and each subset. We also incorporated a site by year interaction for each species. We created a combined model set using the top competing model from each subset of models consisting of all variables of importance to determine effects on nest survival. For modeling DSR prediction, we only included variables (i.e., year, research site, etc.) that met our selection criteria from combined model sets. If treatment effects (ROT/PBG/Pre-treatment) were documented, we ran post hoc analyses to assess potential for within-field variable effects. We calculated the probability of nest success from initiation to fledge (nesting cycle; DSR raised to the power of nesting duration in days for each individual species) to estimate true nest success (Rotella, 2021). Average nest duration in days was based on species-specific nesting information (Cornell Lab of Ornithology, 2019). We present DSR and overall nest success as mean ± SE.

We used resource selection function (RSF; Manly et al., 2002) with a generalized linear mixed model approach with a binomial distribution and a logit link (Bates et al., 2015; Boyce et al., 2002) to examined treatment and within-field variable influences on nest-site selection for facultative grassland birds with large enough sample sizes to allow for proper model performance. We used the glmer function in the lme4 package (Bates et al., 2015) in Program R (version 3.6.2 R Core Team, 2019) to compare RSF for nest sites utilized vs available habitat (i.e., vegetation points sampled in association with point counts) for each pasture across all research locations. We used an unpaired, used vs. unused framework for the RSF analysis (Manly et al., 2002; Milligan, Berkeley & McNew, 2020). This approach allows for a more comprehensive and robust comparison than a nest site paired with a single random point. We followed the previous step-wise modeling approach and model selection criteria described above for DSR. Model subsets for RSF were (1) treatment (ROT, PBG, and Pre-treatment), (2) within-field covariates, and (3) site as a random effect. Significant RSF estimates obtained from the combined model were either a positive score, indicating “use” of a resource in larger proportion than what is available, or a negative RSF score indicating “underuse” concerning available resources (i.e., treatment, within-field variable) (Boyce et al., 2002).

Means and standard errors for all vegetation metrics were calculated for each site, year, and between ROT and PBG pastures. Coefficient of variation (CV) of the vegetation height was estimated to determine structural heterogeneity within pastures and calculated as the standard deviation of the vegetation height divided by the mean × 100 (Bowman, 2001; Chanda et al., 2018; Pearson, 1895).

Results

Daily survival rate and nest success rate

We located and monitored 334 nests across all three sites during the breeding seasons of 2014 –2016. A wide array of facultative grassland bird nests were found, representing a range from one species at QUICK (2016) to 11 at DREC (2014) (Table S1). Three facultative grassland bird species met the selection criteria for data analysis (field sparrow, n = 181; red-winged blackbird (Agelaius phoeniceus), n = 34; and indigo bunting (Passerina cyanea), n = 44). Only two nests of red-winged blackbirds were found in PBG pastures (2015 QUICK and 2016 BGAD) following the 2014 pre-treatment period. Thus, parameters for this species and treatment were inestimable. Estimated VIF values ranged between 1.01 and 1.22, indicating an absence of multicollinearity for all within-field variables for DSR and RSF. Site-by-year interaction models were not incorporated into combined model analysis due to poor model performance (ΔAICc >2.0). Based on top models, DSRs were 0.93 (SE = 0.006) for field sparrow, 0.96 (SE = 0.008) for red-winged blackbird, and 0.92 (SE = 0.01) for indigo bunting. Field sparrow DSR was lowest on PBG pastures and differed from ROT and pre-treatments, while ROT and pre-treatment DSR were similar (Fig. 2). Red-winged blackbird DSR differed among ROT and pre-treatment (Fig. 3). Based on the ΔAICc and combined model DSR beta estimates, ROT and PBG negatively affected red-winged blackbird and field sparrow, respectively (Table 1). Indigo bunting DSR was not influenced by site, treatment, or vegetation metric. Post hoc analysis indicated vegetation height was positively associated with DSR for red-winged blackbird and field sparrow (Table 1). However, the 95% confidence intervals for the β estimate for red-winged blackbirds and field sparrow overlapped zero, indicating a weak effect for vegetation height, yet vegetation height was associated with the top models for each species.

Figure 2 Daily survival rate (DSR) and nest success for field sparrow, red-winged blackbird, and indigo bunting comparing 2 grazing treatments and pre-treatment in the Mid-South, USA, 2014–2016.

Red-winged blackbird DSR for patch-burn grazed pastures were removed due to low sample size (N = 2).

Figure 3 Resource selection function predicted model estimates for field sparrows (FISP), red-winged blackbird (RWBL), and indigo bunting (INBU) for the area used compared with vegetation height (cm).

Vegetation height influenced field sparrow (FISP), red-winged blackbird (RWBL), and indigo bunting (INBU) nest site selection during a patch-burn grazing and rotational grazing study between 3 research sites in Tennessee and Kentucky, USA from 2014–2016.

Table 1 Top-ranked nest survival models (ΔAICc <2.0) and post-hoc sets for top ranked models for selected grassland-associated bird species with support for within-field variables influence on daily survival rates (DSR).

Nests were monitored at three Mid-South sites comparing ungrazed (2014 only) and rotationally and patch-burn grazed pastures, 2015–2016. Model selection was based on Akaike’s information criteria for small sample sizes (AICc), the difference between ranked models (Δ AICc), and model weight or likelihood (ΔAICcwi).

Models	K	AICc	Δ AICc	Δ AICcwi	Variable: β (95% Confidence Interval)	
Field sparrow						
(Combined Model)						
S(∼PBG)	2	491.34	0.00	0.28	PBG: −0.44 (−0.86—0.01)	
S(∼VegHgt)	2	491.38	0.03	0.27	VegHgt: 0.00 (−2.44–0.00)	
S(∼VOR)	2	491.78	0.43	0.22	VOR: 0.00 (−0.00–0.01)	
S(∼PRE)	2	292.98	1.63	0.12	PRE: 0.40 (−0.11–0.92)	
S(∼1)*	1	493.51	2.17	0.09	NA	
Field sparrow							
( Post Hoc )							
S(∼VegHgt+PBG)	3	491.39	0.00	0.24	VegHgt: 0.00 (−0.00–0.00)	PBG: −0.32 (−0.77–0.12)	
S(∼VOR+PBG)	3	491.81	0.42	0.19	VOR: 0.00 (−0.00–0.01)	PBG: −0.32 (−0.78–0.13)	
S(∼NHgt+PBG)	3	492.33	0.94	0.15	NHgt: −0.00 (−0.01–0.00)	PBG: −0.46 (−0.89–0.38)	
S(∼Grass+PBG)	3	492.67	1.27	0.12	Grass: 0.00 (−0.00–0.00)	PBG: −0.37 (−0.82–0.07)	
S(∼Forb+PBG)	3	493.16	1.76	0.10	Forb: −0.00 (−0.01–0.00)	PBG: −0.46 (−0.90–0.02)	
S(∼Lit+PBG)	3	493.35	1.95	0.09	Lit: −0.00 (−0.00–0.00)	PBG: −0.43 (−0.87–0.00)	
S(∼1)*	1	493.51	2.12	0.08			
Indigo bunting						
(Combined Model)						
S(∼1)*	1	140.51	0.00	0.16	NA	
S(∼NHgt)	2	140.85	0.34	0.14	NHgt: 0.00 (−0.00–0.02)	
S(∼VegHgt)	2	140.96	0.45	0.13	VegHgt: 0.00 (−0.00–0.01)	
S(∼BGAD)	2	141.11	0.60	0.12	BGAD: −0.46 (−1.22–0.29)	
S(∼Forb)	2	141.61	1.10	0.09	Forb: −0.00 (−0.01–0.00)	
S(∼Grass)	2	142.24	1.73	0.07	Grass: 0.00 (−0.01–0.02)	
Red-winged blackbird						
(Combined Model)						
S(∼ROT)	2	87.99	0.00	0.28	ROT: −1.09 (−2.20–0.01)	
S(∼DREC)	2	89.47	1.47	0.13	DREC: 0.91 (−0.38–2.22)	
S(∼1)*	1	89.62	1.63	0.12	NA	
S(∼Grass)	2	90.60	2.61	0.07	Grass: 0.00 (−0.00–0.02)	
Red-winged blackbird							
( Post Hoc )							
S(∼VegHgt+ROT)	3	88.71	0.00	0.22	VegHgt: 0.00(−0.02–0.00)	ROT: −1.14 (−2.25–0.02)	
S(∼1)*	1	89.62	0.91	0.14	NA	NA	
S(∼NHgt+ROT)	3	89.67	0.96	0.13	NHgt: 0.00 (−0.01–0.01)	ROT: −1.16 (−2.29–0.03)	
S(∼Forb+ROT)	3	89.78	1.07	0.13	Forb: −0.00 (−0.02–0.01)	ROT: −1.13 (−2.25–0.00)	
S(∼Lit+ROT)	3	89.86	1.15	0.12	Lit: 0.01 (−0.07–0.11)	ROT: −1.19 (−2.39–0.01)	
S(∼Grass+ROT)	3	89.99	1.28	0.11	Grass: 0.00 (−0.01–0.02)	ROT: −1.03 (−2.29–0.22)	
S(∼VOR+ROT)	3	89.99	1.28	0.11	VOR: −0.00 (−0.01–0.01)	ROT: −1.10 (−2.23–0.01)	
Notes.

K is the number of parameters for each model.

VegHgt vegetation height (cm)

INBU indigo bunting

RWBL red-winged blackbird

BGAD and DREC (research sites)

PBG patch-burn grazing treatment

PRE Pre-treatment

ROT rotational grazed treatment

VegHgt vegetation height (cm)

Forb % forb

Lit litter depth (cm)

Grass % grass

VOR visual obstruction reading

NHgt nest height (cm)

* Null model.

Bold estimates indicate significant variables.

Nest success, the overall probability of a nest surviving the nesting cycle (incubation to fledging), was highest for red-winged blackbirds (50% ± 9%, based on 22 ± 5 day nesting cycle) followed by field sparrow (38% ± 5%, 15 ± 10 day nesting cycle), and lowest for indigo bunting (22% ± 0.06%, 19 ± 5 day nesting cycle). The relationship to treatments for nest success was similar to that for DSR for all three species (Fig. 2).

Nest-site selection

We did not find evidence that these three bird species’ nest-site selection was influenced by ROT or PBG treatments. Combined model analysis indicated all three bird species selected nesting locations based on vegetation height and within-field vegetation metrics. Field sparrow nest site selection was positively influenced by vegetation height (β = 0.03, 95% CI = 0.02–0.03; Fig. 3) and negatively impacted by % grass (β = −2.50, 95% CI = −4.36 –−0.64) and % bare ground (β = −2.50, 95% CI = −4.36 –−0.64). Combined RSF model estimates indicated indigo bunting selected nest-sites which had greater vegetation height (β = 0.03, 95% CI = 0.02–0.04; Fig. 3), % forb (β = 0.02, 95% CI = 0.00–0.04) but avoided sites with higher percent grass (β = −0.03, 95% CI = −0.05 –−0.01) and bare ground (β = −0.08, 95% CI = −0.16 –−0.00) (AICc <2.0, ∑ AICc wi = 1.0, Table 2). Red-winged blackbird selected nest sites with greater vegetation height (β = 3.55, 95% CI = 2.77–4.82, Table 2; Fig. 3) but were negatively associated with litter depth (β = −2.50, 95% CI = −4.36 –−0.64) (AICc <2.0, ∑ AICc wi = 1.0, Table 2). A pre- and post-treatment effect was not supported for any species in the combined models indicating a lack of nest-site selection between pre-treatment and treatment pastures (ROT or PBG).

Table 2 Resource selection function results from the top competing model analysis (ΔAICc <2.0) and the closest competing model for nest site selection for three selected grassland-associated bird species.

Nests were monitored at three Mid-South sites comparing ungrazed (2014 only) and rotationally and patch-burn grazed pastures, 2015 - 2016. Model selection was based on Akaike’s information criteria for small sample sizes (AICc), the difference between ranked models (ΔAICc), and model weight or likelihood (ΔAICcwi).

Models	K	AICc	Δ AICc	Δ AICcwi	Variable: β (95% Confidence Interval)	
Field sparrow						
Use ∼VegHgt+Grass+BG+(1—Site)	5	737.38	0.00	0.96	VegHgt: 0.05 (0.00–0.05)	
					Grass: −0.02 (−0.03–−0.00)	
					BG: −0.02 (−0.04–−0.00)	
Use ∼VegHgt+Grass+Forb+(1—Site)	5	743.82	6.44	0.04	VegHgt: 0.03 (0.02–0.03)	
					Grass: −0.02 (−0.02–0.01)	
					Forb: −0.00 (−0.01–0.00)	
Indigo bunting						
Use ∼VegHgt+Grass+Forb+BG+(1—Site)	6	135.88	0.00	1.00	VegHgt: 0.03 (0.02–0.04)	
					Grass: −0.03 (−0.05–−0.01)	
					Forb: 0.02 (0.00–0.04)	
					BG: −0.08 (−0.16–−0.00)	
Use ∼(1—Site)*	3	258.52	122.64	0.00		
Red-winged blackbird						
Use ∼VegHgt+Lit+(1—Site)	4	69.25	0.00	1.00	VegHgt: 3.55(2.27–4.82)	
					Lit: -2.50 (−4.36–−0.64)	
Use ∼PRE+(1—Site)	4	116.98	47.73	0.00	PRE: 23.68 (−4512.75–4560.12)	
Notes.

K is the number of parameters for each model; site was treated as a random effect for each model, (1—site).

PRE pre-treatment (2014)

VegHgt vegetation height (cm)

Forb % forb

Lit litter depth (cm)

Grass % grass

BG % bare ground

* Null model.

Bolding indicates significant variables.

Within-field habitat

A total of 4,464 vegetation samples were collected across all three study sites. Mean vegetation height across all sites and treatments declined following the implementation of ROT and PBG management (Table 3). Sample means for within-field habitat variables differed among sites and treatments (Table S1). Mean vegetation height on each site differed for ROT and PBG as well as between years (Fig. 4). The coefficient of variation for vegetation height varied minimally between ROT and PBG for each site as well as between years (Table 4). Vegetation height maximum and minimum varied for each site and years from a maximum of 225 cm at DREC to a minimum of 0 cm observed at all sites based on random vegetation samples. All VIF estimates for vegetation variables were <5.

Table 3 Total samples collected (N) and means (Standard Error) results for within-field vegetation variables for 3 research sites (BGAD, DREC, and QUICK) across Tennessee and Kentucky.

This data were used to ascertain the impacts of patch-burn grazing and rotational grazing management effects on grassland-associated bird nest-site selection and nest success from 2014–2016.

Site	Year	N	Veg Height (cm)	(SE)	Litter Depth (cm)	(SE)	Grass (%)	(SE)	Forb (%)	(SE)	Litter (%)	(SE)	Bare Ground (%)	(SE)	
BGAD	2014	576	76.30	(1.19)	0.91	(0.07)	83.36	(0.94)	14.27	(0.87)	0.61	(0.18)	0.55	(0.19)	
BGAD	2015	576	29.12	(0.81)	2.20	(0.20)	46.43	(1.15)	14.32	(0.84)	31.97	(1.16)	6.85	(0.73)	
BGAD	2016	576	45.28	(0.74)	0.19	(0.02)	63.81	(1.04)	8.18	(0.65)	19.46	(0.83)	8.19	(0.68)	
DREC	2014	378	70.63	(1.99)	3.33	(0.21)	58.21	(1.76)	3.90	(0.56)	28.20	(1.48)	9.46	(1.01)	
DREC	2015	288	41.94	(1.28)	3.46	(0.18)	47.22	(1.64)	1.58	(0.36)	41.23	(1.89)	9.98	(1.22)	
DREC	2016	306	53.38	(0.87)	2.49	(0.09)	77.04	(0.01)	1.07	(0.00)	19.72	(0.01)	1.58	(0.00)	
QUICK	2014	324	46.01	(1.07)	2.85	(0.22)	56.51	(1.61)	17.55	(1.04)	6.94	(0.58)	18.92	(1.46)	
QUICK	2015	324	25.94	(1.05)	5.28	(0.31)	53.07	(1.56)	5.83	(0.68)	26.34	(1.36)	13.92	(1.34)	
QUICK	2016	342	21.15	(0.77)	0.89	(0.07)	20.07	(0.87)	6.49	(0.58)	50.18	(1.61)	23.41	(1.49)	

Figure 4 Mean vegetation height differences for rotational grazed (ROT) and patch-burn grazed (PBG) pastures during a 3 years (2014–2016).

Research was conducted at BGAD and QUICK in Kentucky, and DREC in Tennessee, USA.

Table 4 Means, standard deviation (SD), and coefficient of variation (CV) for vegetation height on rotational grazed, patch-burn grazed, and pre-treatment pastures at research sites (BGAD, DREC, and QUICK).

This data were used to assess the impacts of each method on grassland bird nest survival and nest-site selection in the Mid-South USA from 2014–2016.

Rotational Grazing (ROT)	Patch-Burn Grazing (PBG)	
Site	Year	Mean	SD	CV	Site	Year	Mean	SD	CV	
	2014*	74.33	28.63	38.51		2014*	77.70	28.62	36.83	
BGAD	2015	29.08	19.33	66.47	BGAD	2015	28.84	19.33	67.02	
	2016	43.29	17.81	41.14		2016	44.53	17.87	40.13	
	2014*	64.85	38.92	60.01		2014*	67.76	38.74	57.17	
DREC	2015	38.92	21.72	55.80	DREC	2015	44.96	21.72	48.30	
	2016	51.11	15.17	29.68		2016	55.80	15.17	27.18	
	2014*	49.05	19.34	39.42		2014*	42.32	19.34	45.69	
QUICK	2015	27.74	18.91	68.16	QUICK	2015	24.65	18.91	76.71	
	2016	22.09	14.28	64.64		2016	20.65	14.28	69.15	

Discussion

Grasslands have been hypothesized to be a significant component of the eastern forests landscape matrix and these areas provided adequate habitat for grassland birds , both obligate (exclusively reliant on grassland habitat) and facultative,over an evolutionary time frame (Askins, 1999). Our research is the first to compare ROT and PBG management effects on facultative grassland bird nesting and nest-site selection in the Mid-South United States and adds to a limited body of work from outside the Great Plains.

We provide evidence that using ROT and/or PBG grazing practices had variable impacts on DSR for three facultative grassland birds. These relationships were also influenced by vegetation height for two of these species. With respect to nest site selection, grazing strategy did not receive support in our models. Instead, birds consistently selected for taller vegetation, regardless of grazing treatments, which reduced vegetation height (except for pre-treatment year), litter depth, and forb cover.

Although DSR was lower for field sparrow on PBG pastures (93%), it was comparable to what has been reported in Pennsylvania (∼93.5%) (Schill & Yahner, 2009). Similarly, red-winged blackbird DSR on ROT pastures was lower (94%) than on pre-treatment year (98%) but was still comparable to DSR reported in the literature (Iowa, ∼96%) (Burhans et al., 2002; Murray & Best, 2003). Spring haying, a disturbance similar to intensive continuous grazing, had an immediate and lasting negative effect on field sparrow and red-winged blackbird (facultative grassland bird) nest survival in Arkansas (Luscier & Thompson, 2009). In Oklahoma however, though field sparrow nests in PBG treatment pastures were fewer than in traditional control grazing pastures, the effects were not significant and overall success (17.6%) was comparable to reported ranges (Holcomb, Davis & Fuhlendorf, 2014). Finally, red-winged blackbird nest densities were greater in idle and cattle exclusion treatment fields, than in those that included experimental disturbances (Lapointe et al., 2003). Even thought we did not document a treatment effect indigo bunting DSR, our results were similar to previous reported DSR (∼93–96%) (Weldon, 2006).

Facultative grassland bird nest site selection has been linked to mean vegetation height across native, restored native, and non-native grasslands in North America (i.e., Illinois, Iowa, West Virginia, and Alberta, Canada) (Fletcher & Koford, 2002; Herkert, 1994; King et al., 2006; Warren & Anderson, 2005). Best (1978) concluded that a reduction in vegetation height in tallgrass prairie systems to <40 cm could result in increased predation risk for some species.

On our pastures, a reduction in vegetation height following habitat disturbance led to reduced DSR for red-winged blackbird and field sparrow (PBG only). Similarly, in Oklahoma, PBG negatively affected field sparrow nest success, which was positively correlated with VOR (Doxon, 2009). It is important to note that due to the incomplete PBG cycle there were sections of 2 years’ worth of growth before the last section was burned. This could have led to favorable environmental conditions for field sparrow nesting and increased vegetation structural diversity across the pasture. Field sparrows prefer undisturbed fields with residual grass (i.e., the previous year’s growth) that provides nest substrate and adequate nesting cover (Best, 1978; Sample, 1989; Sousa, 1983) related to greater DSR. Additionally, field sparrows selected for taller vegetation for nesting as the season progresses (i.e., 27 cm in May –47 cm in July) possibly due to mammalian and snake predation patterns (Best, 1978). Alternatively, but also in the Great Plains, there was no differences in red-winged blackbird DSR between unburned/ungrazed pastures and burned/grazed pastures in a tallgrass prairie (Zimmerman, 1997). In fields burned every 3–4 years, red-winged blackbirds nested in taller vegetation than would be expected, given the height of available vegetation after treatments (King et al., 2006) which has been attributed to predation risk potential from mammalian and reptilian predators (Searcy & Yasukawa, 1995; Picman, Milks & Leptich, 1993).

Indigo bunting DSR was unaffected by ROT or PBG. This species will create nests in old or biennially burned fields, roadside grasses, and woodland edges (Burhans et al., 2002; Payne, 2006). Burhans & Iii (1998) stated that snakes were the principal predator of nests for indigo buntings and vegetation concealment of the nest from below (i.e., reduced vegetation density at ground level) may be the most important factor for this species in Missouri. We believe this could be a plausible explanation for the lack of effect of the vegetation metrics we examine and the low DSR of indigo buntings during our research.

From our results, a reduction in vegetation height following grazing led to reduced DSR for field sparrow (PBG only) and red-winged blackbird, but they continued to select for the tallest vegetation within pastures. For both species, although DSR was lower in PBG pastures, DSR values were similar to those reported from other studies and, for field sparrow, there was no difference between pre-treatment and ROT pastures. Some grassland birds select nesting sites that are infrequently disturbed (1–2 yrs post-disturbance) with greater vegetation structure that could provide for increased concealment and reduce nest predation (Sandercock et al., 2014). Providing a staggered habitat disturbance and low–medium stocking rate across multiple pastures or paddocks could provide grassland nesting birds with an increase in potential nest sites or nesting habitat during the breeding season in the Mid-South USA.

A potential cause of nest mortality during grazing could be failure or abandonment directly caused by livestock (cattle) through trampling or even depredation however, our project did not determine the direct impact of cattle on nest survival (i.e., remote cameras to determine the ultimate cause of nest failure). In native tallgrass prairie in southcentral Canada, nest failures directly attributed to cattle were low (∼0–3%) for various grassland obligate species (Bleho, Koper & Machtans, 2014). In fact, for every nest lost to cattle ∼31 nests were lost to predators (Bleho, Koper & Machtans, 2014). Previous research in Iowa on pastures that had one-third burned annually and low to moderate stocking (1.24–2.97 animal units per month ha−1) exhibited high nest survival for eastern meadowlarks during the first year of a 2-year study (Hovick & Miller, 2016). Additionally, paddocks lightly grazed by cattle (i.e., 15 cattle/5 day in a 2 ha paddock) reduced nest abandonment or failure caused by cattle (Campomizzi et al., 2019). However, based on previous research, we are confident stocking rates were light enough (2.5–5.0 ha−1) to minimize nest failure or abandonment caused by cattle within the Mid-South USA.

Our research provides the first experimental use of ROT and PBG on NWSG pastures and the effects on facultative grassland bird DSR and nest-site selection in the Mid-South USA. Previous research on ROT and PBG management effects on grassland bird breeding dynamics has been conducted in the Mid-West USA where tracts of managed lands are much larger (i.e., ∼5,000–18,000 ha pastures) and under arid climatic (i.e., ∼31 cm of precipitation) conditions. Researchers have cautioned about extrapolating habitat or landscape effects for a wide-ranging species (i.e., field sparrows) and across ecosystems (Winter et al., 2006). Current grazing practices on pastures in the eastern USA involve year-round stocking, mowing, or hay production due to the higher precipitation and longer growing season than the western ecoregions (Askins et al., 2007; Monroe, Burger & Martin, 2019; Warren & Anderson, 2005). Restoring Mid-South pastures currently dominated by exotic cool-season grasses to NWSG may be accelerated by adoption of grazing methods such as PBG from the semi-arid Great Plains (Keyser et al., 2019). Additionally, if large tracks of pasturelands across the Mid-South return to NWSG, grassland birds could benefit from an increase in potential nesting habitat (West et al., 2016). Yet, until significant pasturelands of the Mid-South are restored to NWSG, we have provided baseline information by comparing ROT and PBG management to NWSG in the Mid-South for grassland bird reproductive efforts and nest-site selection that could guide conservation strategies and future research.

Conclusions

Due to the extreme decline in grassland bird populations, it is imperative to fully explore alternative livestock production strategies and their impacts on grassland bird populations across ecoregions outside of the Great Plains. Our research shows that ROT and PBG management of NWSG can have variable impacts on nesting success but little direct impact on nest-site selection for facultative grassland bird species. Geller, Sample & Henderson (2004), Powell (2006), and Weir et al. (2013) state that ∼2.5–4 years following patch-burns can allow vegetation biomass and litter to accumulate which can provide adequate nesting cover for birds that utilize ground and standing vegetation to create nests. Our short-term data for the 2-year post-treatment period provided some support for nest site selection for field sparrows. Based on this information, it is important to consider trade-offs between habitat disturbances and potential short-term impacts on grassland breeding birds. Additionally, our research highlights the importance of continued monitoring, because we do not know how our pastures will continue to change and species respond over longer time intervals. It is also important to note that our PBG treatment cycle (i.e., all three sections burned) had not been completed by the end of the study yet previous research has shown PBG can be useful in creating habitat disturbance for grassland birds (Churchwell et al., 2007; Coppedge et al., 2008; Fuhlendorf et al., 2006). With a lack of disturbance, grassland ecosystems in the Mid-South USA will quickly progress to later seral stages, thereby reducing available breeding habitat for grassland obligate bird species which can further exacerbate population declines.

Grazed native grasses appear to offer the opportunity to maintain livestock production while simultaneously achieving grassland bird conservation goals (Allred et al., 2014; Fuhlendorf et al., 2006). Managers can utilize ROT, PBG, and unburned/ungrazed areas in a rotation mosaic of vegetation that differs by age and size. Creating such a mosaic can create a heterogeneous vegetation structure that enhances grassland bird nesting habitat and nesting species diversity (Delany & Linda, 1998; Fuhlendorf & Engle, 2004; Hovick et al., 2015; Monroe et al., 2016). Our research along with Campomizzi et al. (2019) indicate that some combination of PBG, ROT, and unburned-ungrazed areas (i.e., our pre-treatment year, 2014) can provide adequate nesting habitat on small pasture lands (∼1.8–7.8 ha) for a variety of grassland birds. We encourage future research to monitor nesting survival with cameras to determine the ultimate cause of mortality.

Supplemental Information

Supplemental Information 1 Nesting and random habitat samples for nest site selection analysis

Click here for additional data file.

Data S1 Nest survival data set

Nesting data and vegetation measurements collected for nest survival.

Click here for additional data file.

Table S1 Grassland bird nests found on rotational grazed (ROT), patch-burn grazed (PBG), and pre-treatment (Pre) pastures at 3 different research sites (BGAD, DREC, and QUICK) to assess the impacts of each method on nest survival and nest-site selection in the Mid

Click here for additional data file.

We greatly appreciate D. Ditsch and his staff at University of Kentucky’s Robinson Center for Appalachian Resource Sustainability and T. Keene from the University of Kentucky’s Department of Plant and Soil Sciences for his efforts in the success of this project. We also thank H. Moorehead and K. Thompson and their staff for their outstanding work at DREC, and T. Edwards and M. Schroder for all the hard work at BGAD. Also, we thank the many field technicians for the devotion and attention to detail on this project. We would like to thank the anonymous reviews and academic editor.

Additional Information and Declarations

Competing Interests

Author Contributions

Animal Ethics

Field Study Permissions

Data Availability

The authors declare there are no competing interests.

Byron R. Buckley analyzed the data, prepared figures and/or tables, authored or reviewed drafts of the article, and approved the final draft.

Christopher M. Lituma conceived and designed the experiments, performed the experiments, analyzed the data, prepared figures and/or tables, authored or reviewed drafts of the article, and approved the final draft.

Patrick D. Keyser conceived and designed the experiments, performed the experiments, analyzed the data, prepared figures and/or tables, authored or reviewed drafts of the article, and approved the final draft.

Elizabeth D. Holcomb conceived and designed the experiments, performed the experiments, authored or reviewed drafts of the article, and approved the final draft.

Ray Smith conceived and designed the experiments, performed the experiments, authored or reviewed drafts of the article, and approved the final draft.

John J. Morgan conceived and designed the experiments, performed the experiments, authored or reviewed drafts of the article, and approved the final draft.

Roger D. Applegate conceived and designed the experiments, authored or reviewed drafts of the article, and approved the final draft.

The following information was supplied relating to ethical approvals (i.e., approving body and any reference numbers):

University of Tennessee - Institutional Animal Care and Use.

The following information was supplied relating to field study approvals (i.e., approving body and any reference numbers):

Blue Grass Army Depot (BGAD)

T. Edwards (United States Army)

Dairy Research and Education Center (DREC)

H. Moorehead and K. Thompson (University of Tennessee - Knoxville)

Robinson Center for Appalachian Resources Sustainability (QUICK)

D. Ditsch and T. Keene (University of Kentucky)

The following information was supplied regarding data availability:

The raw files are available in the Supplementary Files.

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
