# Peer review of "Effects of grazing strategy on facultative grassland bird nesting on native grassland pastures of the Mid-South USA"

_PeerJ, doi:10.7717/peerj.13968_

## Round 0.1 · original submission · Minor Revisions

Both reviewers found your manuscript to be a solid study needing mainly points of clarification. They have provided detailed suggestions for consideration. In your rebuttal, please include the comments and questions that Reviewer 1 included in the attached manuscript as well as those on the report because these concern substantial issues, not simply style or grammar.

·

Basic reporting

Dear Editor I have reviewed the paper entitle “Effects of grazing strategy on grassland bird nesting on native grassland pastures of the Mid-South USA (#73034)” which provided a baseline information by comparing rotational grazing (ROT) and patch-burn grazing (PBG) management to native warm season grasses (NWSG) in the Mid-South for grassland bird reproductive efforts and nest-site selection with relevance for management and conservation strategies of pastures. The work is well written and generally clear. My main concern was within “material and methods” and “results” sections which were a bit difficult to read. The statistical analysis section needs more clarification, I have made some comments directly in the manuscript that should be attended in order to improve understanding of the work. I think that the paper is correct and the authors just need to add a bit more information. Also the discussion can be shortened.
I enjoyed reading this manuscript! I attach the manuscript with my comments

Experimental design

I think experimental design is correct and consistent with the objectives, although few clarifications are needed (L. 139-146, 171 and 177).

Validity of the findings

The work is solid, interesting but most of all, necessary!

Additional comments

Statistical analysis section should have more information such as the response variables, structure of statistical models and the corresponding justification.

Reviewer 2 ·

Basic reporting

This is a robust study that merits publication but requires some additional detail and revised framing to make it more meaningful and powerful, to better showcase what it adds to the body of studies on responses of grassland birds to management. Basic reporting is generally fine. Some editing is needed and I make some suggestions for additional literature references to include and also suggest reframing hypotheses/results reporting to better distinguish them from what is already known about these species' habitat requirements and their responses to grassland management. General comments below:


I think the paper could be substantially improved by adding a more detailed exploration of the habitat requirements and responses of the three focal species. I do not believe it is very meaningful to have a general hypothesis for all species simply because they have different needs and patterns; therefore I suggest distinguishing them according to their habitat requirements and discussing them accordingly. This would involve some work to reframe the paper and update the literature review (in minor, not major, ways) but I think it would make the paper more clear and powerful and more useful to researchers and managers.
I would also add a paragraph in the introduction with brief information about the three focal species and why they were chosen, and perhaps a brief discussion of this suite of grassland birds. It would be helpful to restate your hypotheses/predictions in terms of these species’ life history characteristics and what is already known about them in terms of their responses to grazing, burning, vegetation height, etc.

The three study species (FISP, RWBL, INBU) are all associated with grasslands, but I noted that all are distinguished from “obligate” grassland species like BOBO and GRSP (among others) in that FISP and INBU are require edges/shrubs/woody vegetation for nesting. I do not wish to split hairs, but these species will naturally respond differently to disturbance than “obligate” grassland species, in the sense that they require shrubs/field edges for nesting (and in some cases actually decline with increasing grass cover, as the authors found). By contrast, BOBO and GRSP nest in large expanses of grassland and often may decline with woody vegetation. RWBL also require wetlands of course. I suggest the authors simply indicate their awareness of this in the paper and ideally also the abstract by referencing the use of woody vegetation, perhaps simply by introducing them as facultative (vs obligate) or any other reasonable terminology to capture this. In my own work on “obligate” grassland birds we did not have a significant presence of FISP or INBU except in fields/riparian areas lots of woody vegetation (and/or wetlands in the case of RWBL). This is not intended as a criticism but simply a point I think is useful and important for those working to conserve grassland bird species. The distinction between obligate and facultative grassland species is discussed in Zimmerman 1992 (Density-independent factors affecting the avian diversity of the tallgrass prairie community. Wils. Bull. 104, 85–94) and I suggest the authors include a recent reference to this as well.
Comments to various specific lines in sections are below. Well done by the authors. Thank you for the opportunity to review this paper. I hope these comments are helpful.

Abstract –

General question/suggestions:
1 - Is it possible to be a bit more specific in the abstract about HOW PBG and veg height affect FISP and about how ROT plus veg height are important for RWBL? This would be helpful for readers who may only read the abstract, and to give an idea of what your study adds to the literature. Most people who study grassland bird responses understand that veg height and disturbance affect them, but the HOW is what will provide new information for the study species, so I would recommend adding a sentence or two to detail these effects if possible.

2 – I suggest perhaps adding a paragraph to qualify differences between suites of bird species with different life histories, which are on a gradient.

Just for reference, and to better explore these species’ habitat requirements and responses, I went to Birds of the World (accessed via AOS) and have copied and pasted these species’ profiles below:
“The Field Sparrow is a common songbird of eastern North America, breeding in brushy pastures and second-growth scrub, but avoiding similar habitat in developed areas… Generally in successional old fields, woodland openings and edges, roadsides and railroads near open fields. Does not breed close to human habitation; occasionally found in Christmas tree farms, orchards, and nurseries (Peterjohn and Rice 1991). Will nest in old fields directly after a burn or within a year of cultivation, but only if there is scattered woody vegetation with elevated perches in the territory. As thickets of trees spread in the habitat, numbers decline. The general trend for old field habitats is that Field Sparrows begin breeding within 1-2 years after human uses stop; population sizes rise for perhaps a decade, then decline. After ~30 yr of old field succession, the habitat is overgrown with trees and shrubs and no longer used for breeding (MC).”
The Red-winged Blackbird is widely distributed, breeding in open wetland and upland habitats… Red-winged Blackbirds breed in a variety of wetland and upland habitats. Wetland habitats include freshwater marsh, saltwater marsh, and rice (Oryza sativa) paddies. Upland breeding habitats commonly include sedge (Carex spp.) meadows, alfalfa (Medicago sativa) fields and other crop lands, and old fields; breeds less commonly in wooded areas along waterways and in open patches in woodlands (2, 7, 6). Roosts during breeding season in habitats with dense cover, especially in wetlands (76, 77).
Indigo Buntings live in shrubby areas and weedy fields. Their colorful appearance and cheerful songs are good reasons to fallow old fields and to spare (not spray) herbicides along railways and roads… In Michigan, shrubby and weedy habitats between woods and field, thickets, shrubby swamps, upland areas of old fields, upland woods and mesic woods of sugar maple (Acer saccharum) (Payne 1989, Payne and Payne 1998). Scarce in wooded swamps with tamarack (Larix laricina); when these woods were decimated in 1990s by larvae of larch casebearer moth (Coleophora laricella), buntings held territories in more open sites where not seen in earlier years (RBP).
Common in fields in vegetational succession from cultivation to shrubby habitat, establishing territories only in sixth and seventh years after cultivation and returning to those ares for > 30 yr; appear to respond to presence of dense, upright woody vegetations for song perches and cover (Lanyon 1981). In forested West Virginia, more numerous in shrubby sites than in sapling sites; found in canopy gaps where trees defoliated by gypsy moths (Lymantria dispar) (Bell and Whitmore 2000). In managed forest in Missouri, appeared in cleared habitats the year after forests were cut, and were abundant 3-4 yr afterwards (Kabrick et al. 2004). In Arkansas where plots of different sizes were cleared 6-7 yr earlier, Indigo Buntings settled and bred both in large plots and in shrubby clearings < 0.4 ha in area (Alterman et al. 2005). In dry western Great Plains, found in wooded floodplains and ravines; in isolated populations in southwest U.S., found in brushy canyons and wild rose. In much of range, seek out edges of woods and fields, cut-over woodlands, abandoned fields and roadsides a few years after lands are cleared (eg. Suarez and Robinson 1997, Yahner et al. 2002).
Generally absent in urban and suburban areas, in intensively cultivated and grazed areas, in closed-canopy forests, and in deserts except around shrubs near waer (Sutton 1959, Taber and Johnston 1968, Wauer 1997).

In western North America, habitat much as in Lazuli Bunting along wooded rivers in Great Plains (Sibley and Short 1959b, Sutton 1967b, Emlen et al. 1975, Baker and Boylan 1999 . In Utah in Zion National Park, both species are in brushy side canyons, the Indigo Buntings in brushy vegetation and Lazuli Buntings on flood plains along river (Wauer 1997).

>>> END OF BOTW EXCERPTS >>>
>>>

Specific note:
Line 29 – there is a missing “for” after the word “important”

Introduction

Specific notes:

Line 54: Can you additional, including more recent, references to support this statement?

Lines 67-68: Regarding the fact that studies have reported highly variable nest success in response to ROT, this is not surprising in the sense that bird responses differ by species and life history (group versus shrub nesting species). As already mentioned, there is a suite of “obligate” grassland nesting birds versus these study species. Thus I feel this point is not very meaningful, unless you are focusing on similar species to your study species (or previous studies of the same species). Is this the case? Regardless, I think it is appropriate to point out the connection between specific habitat requirements for nesting and known responses or expected responses to disturbance in that context.

Line 72: I would qualify this statement by indicating that “pyric-herbivory” is a hypothesis in the sense that we were not there to measure historical natural disturbances during the development of North American grasslands; I believe it is very likely to be the case, but nevertheless a belief as opposed to a measurable/provable fact. Therefore I would indicate this by stating that pyric-herbivory is believed/hypothesized to mimic historical natural disturbances…

Experimental design

Generally OK; I had a few questions and requests for clarification:

Methods

Lines 169-170 Regarding determining fledging, would “adult alarm call” alone be used as evidence of fledging, without accompanying evidence (visual confirmation, chick feeding calls)? I do not believe this is sufficient if so, but please revise the text to clarify this.

Lines 189-190 “we selected nests of those species that nest in grasslands or pasturelands” seems superfluous because the purpose of the study is to investigate grassland birds and the study sites were in grasslands. Or were there other nesting birds in your study sites that were eliminated because they were not grassland birds? Please revise to clarify

Validity of the findings

I suggest some improvements to emphasize the novelty of the findings. Specific notes below:

Results

Did you have any obligate grassland birds, as referenced earlier in this review? It might be of interest to list all species together with numbers of nests as a supplementary table for interested researchers.

Lines 268-269 “None of these three bird species’ nest-site selection was influence by…” I would rephrase this to state you did not find evidence that these three bird species’ nest site selection was influenced, since that is more accurate!

Lines 269-270 “all three bird species selected nesting locations based on vegetation height and within-field vegetation metrics” – what metrics specifically? Without mentioning them, this phrase is not very meaningful… if specifics are too complicated to mention here, indicate they will follow below.

Lines 270-278: FISP was reported as “positively influenced by vegetation height” whereas both INBU and RWBL are described as influenced by vegetation height without a verbal indication of how they are so influenced. As stated above, such statements are not very meaningful without additional detail.
It is also interesting, although these birds are indicated in the title and throughout the paper as grassland birds, that FISP and INBU are described as negatively influenced by % grass. This reinforces the point that these two species are associate with woody/edge vegetation, in contrast to obligate grassland species like BOBO and GRSP. Some additional details about the factors associated with nest-site selection would make this paragraph more meaningful, and demonstrate more effectively that this study brings something new to the table.

Discussion

Lines 340-341: The authors mention that the contrasting response of INBU to treatments highlights the value of including multiple species when assessing grazing management impacts on birds. Is this actually the case? While we would certainly expect a diversity of responses from a diversity of species, if the purpose of a study is to ascertain such patterns, and not enough information or consideration is given to understanding species’ responses based on their life history characteristics, one can also argue that including multiple species without in-depth treatment of any of them can simply dilute and confuse the findings. The authors need to make a stronger case for why and how it is valuable for multiple species to be considered at the same time, in my view, or drop this statement.

Lines 343-347: This is an interesting and plausible explanation of INBU responses; bringing more attention to other possible explanations for the responses of all three species would make the paper more meaningful and interesting and provide a framework for follow-up research.

Lines 358-371: This is an interesting paragraph that I think could be improved with a greater focus on stocking rates/grazing intensity. It is not very meaningful to discuss direct impacts of grazers without reference to stocking rate since stocking rate can make the difference between high and low nest survival. The authors do mention stocking rate in the final sentence of this paragraph; I suggest it be moved closer to the beginning of this paragraph (or start a subsequent paragraph) and followed by a brief discussion of stocking rate as it impacts birds; see for example Kaplan et al. 2021 regarding negative impacts of high densities of grazing bison on Bobolink abundance and nesting and contrasting findings by Boyce et al. 2021 on other species with much lower stocking rates of bison. In my view it is not enough to say you are “confident” your stocking rates were light enough to minimize direct impacts of cattle without providing a quantitative basis for this – you could do so by comparing to the above studies, those you already cited, or others.

The authors briefly mention deferred grazing (line 368) but without reference to their own study system; please comment on this in your own study system (whether or not deferred grazing was used) and you might consider adding a comment about this in the conclusion if appropriate.

Finally, given the distinction between the focal species and "obligate" grassland species, would you recommend similar research also focus on such species? Please comment.

Additional comments

My comments appear in the above 3 sections.

---

## Round 0.2 · Minor Revisions

I have read your revised manuscript and response to the reviewers' comments. I consider that the changes made are adequate without the need to send the manuscript out for a second review. The manuscript is sometimes a challenging read with so many unfamiliar acronyms to remember (at least, to me). However, they all seem to be well used, so they should probably stay. Supp. Table 1 should include scientific names under each of the common names. The manuscript does contain quite a few minor misspellings, missing words, extra words, punctuation errors, etc. I have highlighted these on the attached pdf with a comment to indicate the suggested replacement. I was not always certain that I understood your intended meaning, so make the changes with care. Remarkably, compared to most manuscripts I edit, I found no errors in the reference details and formatting. Thanks for your attention to detail.

---

## Round 0.3 · accepted · Accept

The authors have made the required changes.